# Risky Behaviour among Nurses in Poland: An Analysis of Nurses’ Physical Condition, Mental Health, and Resilience

**DOI:** 10.3390/ijerph18041807

**Published:** 2021-02-12

**Authors:** Lucyna Gieniusz-Wojczyk, Józefa Dąbek, Halina Kulik

**Affiliations:** 1Department of Propaedeutics of Nursing, School of Health Sciences in Katowice, Medical University of Silesia in Katowice, 20/24 Francuska Street, 40027 Katowice, Poland; hkulik@sum.edu.pl; 2Department of Cardiology, School of Health Sciences in Katowice, Medical University of Silesia in Katowice, 45/47 Ziołowa Street, 40635 Katowice, Poland; jdabek@sum.edu.pl

**Keywords:** nurse, resilience, wellbeing, health, smoking, alcohol abuse, eating habits

## Abstract

*Background*: Nursing is a profession where staff are exposed to chronic stress. Mental resilience plays a significant role in the process of coping with these challenges. The aim of this study was to assess nurses’ mental and physical wellbeing, as well as resilience, by taking into account the occurrence of risky behaviour among nurses in Poland. *Methods*: A descriptive study was carried out between June 2017 and May 2018 among nurses (*n* = 1080) employed in primary healthcare or in training centres in Silesia, Poland. Data were obtained from a number of questionnaires. *Results*: Over half of the nurses (*n* = 735; 68%) had an average psychophysical mood level resulting from stress, and 179 (16.6%) nurses had a low psychophysical mood. Those with a lower psychophysical mood showed a greater tendency towards developing improper eating habits (*r* = −0.23; *p* < 0.001). Most nurses had an average (*n* = 649; 60.1%) or low (*n* = 255; 23.6%) level of resilience. Higher resilience levels were observed in nurses aged over 30 years (*p* = 0.004) and in those with additional employment (*p* = 0.008). High resilience was associated with a lower intensity of risky behaviour. *Conclusion*: Most nurses in Poland display average and low levels of resilience, which can have unfavourable consequences for their health.

## 1. Introduction

The current global nursing shortages have become a global challenge for health organisations, clinicians, scientists, and nursing teachers [1]. Nurses are the largest professional group among healthcare professionals [2], and, in comparison to healthcare professionals in general, they are the most exposed to stress [3]. Improper stress management can have a negative impact on nurses’ mood, resulting in depression, dissatisfaction with their job, reduced organisational loyalty, and planning to quit work, consequently leading to professional burnout and/or various diseases [4,5,6,7]. Experiencing strong or chronic stress may also contribute to an increase in risky behaviour (e.g., alcohol abuse, smoking, and poor diet). Studies show that nurses in many countries consume more alcohol [8], smoke more [9], and are more obese in comparison to the population in general [8] or employees from other sectors [10]. Such poor lifestyle choices not only affect nurses’ professional performance but also increase the burden on our healthcare system since these are all key risk factors for the development of chronic diseases, such as cardiovascular diseases, hypertension, or type 2 diabetes [11,12]. Resilience is recognised as the personal capacity to effectively adapt to difficult situations. It can be defined as a personality trait or a dynamic process. The common denominator of various attitudes consists of the assessment of resilience as a property which enables people to maintain an optimum level of effective performance and to cope with failures quickly and easily, despite the difficulties encountered [13], which can be considered an alternative method of alleviating hardship in the workplace. Resilience is also perceived as an important pro-health resource [14,15]. High resilience is conducive to experiencing more positive emotions, such as gratitude, interest, or love, and a lower intensity of negative emotionality (anger, sadness, or fear) [16]. Resilience as a personal resource is used in professional situations [17]. Resilience can protect an individual against stress [18]. Bearing in mind that stress often leads to using stimulants in order to alleviate its effects, it can be assumed that resilience will protect nurses against excessive preoccupation with eating, excessive alcohol consumption, and smoking. Bad habits have a special place among the ways of coping with stress. They mainly serve the function of regulating one’s emotional state [19]. According to the Lazarus and Folkman concept, coping with stress is a constantly evolving cognitive and behavioural effort directed to specific external and/or internal requirements that are assessed as aggravating or exceeding human capabilities [20]. So far, little research has focused on the influence of personality factors (resilience) on risky behaviour. This is undoubtedly one of the first reports showing how the features of mental resilience can influence the behaviour of nurses in Poland.

This study aimed to assess the psychophysical wellbeing and level of resilience among Polish nurses. We also explored the associations between wellbeing and resilience levels and the occurrence of risky behaviour in nurses, such as smoking, excessive alcohol consumption, and poor diet.

## 2. Materials and Methods

### 2.1. Design and Participants

This was a descriptive, multicentre study carried out between June 2017 and May 2018 in the following healthcare units and training centres for nurses in Silesia: Provincial Specialist Hospital no. 4 in Bytom, The Kornel Gibiński University Teaching Centre at Silesian Medical University in Katowice, Nursing and Midwifery Training Centre in Łagiewniki, and Postgraduate Nursing and Midwifery Training Centre in Krosno. A total of 1064 nurses were included in the study on the basis of an estimation of the size of the nursing population in Poland (*n* = 288,395) (Supreme Chamber of Nurses and Midwives, 2017) (the size of fraction = 0.5; the level of confidence = 95%; maximum error = 3%).

All nurses were informed about the purpose and nature of the research, as well as the application of the obtained results. All participants consciously and voluntarily gave their consent to participate in the research. The research included nurses holding a diploma with the professional title of nurse who had been in the profession for at least 1 year. Pregnant women, those who had practised for less than 1 year, and those who did not give their informed consent were excluded.

The Bioethical Committee at the Silesian Medical University in Katowice approved the study (KNW 0022/KB/89/1/17). All participants gave their consent to be included in the study by returning completed, anonymous questionnaires.

After meeting the aforementioned criteria, participants were provided with the survey in an envelope. After completing the survey, the respondent returned it to the researcher.

The questionnaire contained questions concerning demographic data (age, sex, length of service, type of ward, marital status, and additional employment). It also contained a Polish version of the AUDIT-C test (the Alcohol Use Disorders Identification Test for Consumption; https://auditscreen.org (accessed on 12 December 2020)) to identify alcoholism [21]. In particular, women who consume ≥2 units at a time or men who consume >4 units at a time are considered at risk of harmful drinking [21,22,23]. The study also included the Polish adaptation of the Fagerström test for nicotine addiction for the assessment of smoking rates (Cronbach’s alpha of 0.820). The Fagerström test evaluates pharmacological addiction to nicotine. Collecting 7 or more points indicated that the individual is probably pharmacologically addicted to nicotine, while getting below 7 points means that smoking is a learned behaviour [24,25]. It also contained the questionnaire “My Eating Habits” (MEH) by Nina Ogińska-Bulik and Leszek Putyński (Cronbach’s alpha of 0.89) [26] which consisted of 30 statements. Each diagnostic answer scores 1 point. The total number of points enables one to determine a general tendency for improper eating habits (a high score obtained in the questionnaire points to improper eating habits, i.e., the tendency to overeat or abstain from eating). Due to the nature of overeating, three factors are distinguished, each of which contains 10 questions: habit overeating (0–10 points), emotional overeating (0–10 points), and the tendency to restrain from eating (0–10 points). This tool enables the diagnosis of eating disorders, predicts the tendency for being overweight, and is used in the selection of interventions, the aim of which is to reduce excessive body weight [26].

Wellbeing was measured using the Psychosocial Working Conditions (originally PWP) (Cronbach’s alpha of 0.90) questionnaire and two factors related to physical and mental dimensions, which are collectively referred to as mental and physical conditions. The D1 scale consists of a general assessment of physical health and stress, as well as the occurrence of somatic symptoms, such as headaches and stomach and heart problems. Factors concerning mental wellbeing (D2) focus on the assessment of negative emotional states, life and job satisfaction, and self-confidence. High values reflect a higher level of wellbeing. The questionnaire contains standards developed for eight professional groups, including nurses. The results can be expressed as a sten score, where results in the range 1–4 mean low wellbeing, those in the range 5–6 mean average wellbeing, and those in the range 7–10 mean high wellbeing [27]. Lastly, we used the Assessment of Resiliency Scale (SPP-25) by Ogińska-Bulik and Juczyński (Cronbach’s alpha of 0.89) [28] to measure the overall level of resilience among the nurses. The scale measures the five constituting factors of resilience: (1) determination and persistence in action, (2) openness to new experiences and a sense of humour, (3) personal competencies to cope and tolerance of negative effects, (4) tolerance of failures and treating life as a challenge, and (5) an optimistic life attitude and ability to mobilise in difficult situations. These five characteristics are rated on a five-point Likert scale (from 0—strongly disagree to 4—strongly agree). The overall score on the SPP-25 is the sum of the five aforementioned factors (i.e., five points per item). A higher score denotes a higher level of resilience. The overall result of SPP-25 can be expressed on a sten scale, in which the results ranging from 1 to 4 mean low resilience, those ranging from 5–6 mean average resilience, and those ranging from 7–10 mean high resilience [28].

Improperly completed questionnaires were excluded from the analysis.

### 2.2. Statistical Analysis

As far as quantitative variables are concerned, the Mann–Whitney U test was used to assess differences between the two groups. The chi-squared test of independence was used to assess dependencies between variables in the nominal and ordinal scale. Some risk factors were compared using the Kruskal–Wallis analysis of variance (ANOVA) test or a one-way ANOVA. Spearman’s correlation coefficients were used to evaluate the correlations between quantitative variables. All statistic tests were calculated at the significance level alpha ≤ 0.05 and performed using SPSS and Statistica software.

## 3. Results

### 3.1. Characteristics of Studied Group

In total, 1200 participants met the inclusion criteria and were provided with the survey in an envelope. Of them, 1080 returned the completed survey (i.e., a response rate of 90%). The nurses subject to the analysis were aged 24–63 years (the mean age was 42.8 years). Nearly half (44%) had worked in their profession for over 20 years (*n* = 484), and almost 40% (*n* = 397) held additional employment. Overall, 379 (35%) nurses consumed alcohol in a harmful way, and ~20% were smokers.

The general characteristics of the study group are summarised in Table 1.

### 3.2. Psychophysical Wellbeing

The mental and physical condition of the group of nurses under analysis was determined with the use of the wellbeing scale (D) of the Psychosocial Working Conditions questionnaire [27]. The overriding question in the theoretical scale (D) was the following: “How do you feel?” The D1 scale involves an overall assessment of physical health and stress and the occurrence of somatic symptoms, such as headaches and stomach and heart problems. The mental wellbeing (D2) scale focuses on the assessment of negative emotional states, life and job satisfaction, and self-confidence.

Over half of nurses (*n* = 735; 68%) had an average psychophysical mood according to the PWP scale, and 179 (16.6%) had low mood levels (Table 1).

Widows or widowers were found to have a lower psychophysical mood (mean [standard deviation] score of 3.31 [0.64] on the PWP scale) than married nurses (3.58 [0.52]) or those who were single (3.61 [0.48]; F = −3.035; *p* = 0.028; one-way ANOVA). Moreover, nurses who were additionally employment had a better sense of wellbeing (mean score of 3.62 [0.52]) than those without additional employment (mean score of 3.54 [0.52]; Z = −2.395; *p* = 0.017; Mann–Whitney U test; Table 2).

The correlations between psychophysical wellbeing and nicotine dependence, alcohol consumption, and eating habits are presented in Table 3.

### 3.3. Resilience

We used the SPP-25 scale to evaluate nurses’ mental resilience [28].

Nurses with an average level of resilience predominated (*n* = 649; 60.1%), followed by those with low (*n* = 255; 23.6%) and high resilience levels (*n* = 176; 16.3%).

Younger nurses (<31 years of age) had lower resilience scores than those aged over 30 years old (*p* = 0.004, Kruskal–Wallis ANOVA). Nurses with additional employment displayed higher levels of resilience (mean score of 69.46 [14.49]) than individuals without additional employment (mean score of 67.46 [15.01]; Z = −2.656; *p* = 0.008; Mann–Whitney U test; Table 4).

Nurses with high levels of resilience had a significantly better psychophysical mood (mean score of 3.80 [0.53] on the PWP scale) than those with low (mean score of 3.28 [0.53]) and average (mean score of 3.62 [0.46]) levels of resilience (*p* < 0.01; Kruskal–Wallis ANOVA).

Nurses with high resilience had a significantly lower risk of alcohol addiction (*p* = 0.003; chi-square test) and better eating habits (overall score on the MEH scale; *p* = 0.001; Kruskal–Wallis ANOVA) than individuals with average and low resilience levels (Table 5).

## 4. Discussion

Several studies have shown that nurses are burdened with job-related stress [29,30,31,32,33]. In this study, we determined the effects on the health of nurses using the wellbeing scale (D) of the PWP questionnaire [27], which enables an assessment of their physical and mental mood. We found over half of the nurses in this study had an average psychophysical mood and 17% had low mood levels (note: participants scored higher on the physical wellbeing scale than on the mental mood scale). Our findings reflect those reported among nurses working in another region of Poland who had similar mean scores obtained on all scales of the PWP questionnaire [34,35]. Previous multicentre studies have found common prognostic factors affecting psychophysical mood including age, job satisfaction, sleeping disorders, years of employment, and marital status [34,35,36,37,38]. We found that psychophysical mood was also influenced by marital status, as well as additional employment (i.e., nurses who took on additional employment had better wellbeing), which suggests that wellbeing may impact nurses’ willingness to work. This finding concurs with studies conducted in 2002–2005 with the international research programme called the NEXT Study (Nurses’ Early Exit Study) aimed at identifying the reasons why nurses leave the profession. In particular, measures of mental wellbeing were strongly related to the intention of “leaving” a job and may largely determine the actual decision to resign [39].

Stress and the inability to cope can have a negative impact on employees’ wellbeing, thus leading to burnout and/or the development of various diseases [6,40,41]. Moreover, high levels of stress can result in or aggravate non-adaptation behaviours, such as smoking, overeating/undereating [42,43], excessive alcohol consumption, and substance abuse. However, we found that nurses’ psychophysical mood was only weakly correlated (*r* < 0.2; *p* = 0.001) with smoking (assessed using the Fagerström test for nicotine dependence) and risky alcohol consumption (as evaluated by AUDIT-C). A slightly higher result (*r* = −0.23; *p* = 0.001) was observed between psychophysical mood and scores on the My Eating Habits questionnaire, indicating that nurses with low psychophysical mood may have a greater tendency to have improper eating habits.

The mechanism of self-regulation, protecting nurses against the negative consequences of experiencing both traumatic and everyday events, consists of resilience [44]. We found that Polish nurses experienced average levels of resilience, which is in line with many national and international studies [13,17,45,46], as well as people working in the Polish normalisation group [28]. As in the previous studies among nurses [17,47,48], we found that younger nurses (<30 years) had lower resilience than older nurses. The findings indicate that there is a great need for healthcare organisations and nursing leaders to develop programmes that focus on building better resilience among younger and less experienced nurses. Furthermore, we demonstrated that nurses with additional employment displayed higher levels of resilience than those who did not take up additional employment. This finding supports previous research suggesting that individuals with higher resilience are more open to new experiences and engage in relationships at greater levels [49]. We also found that nurses with a high resilience level had a significantly better psychophysical mood (reflecting the perceived stress level) than those with low and average resilience, as observed in earlier studies [44,50,51]. Resilience is also treated as an important resource that protects against professional burnout [52] and the development of post-traumatic stress disorder (PTSD) symptoms [53]. Moreover, the lowest number of nurses consuming alcohol in a harmful way was found among individuals possessing high levels of resilience. These results are consistent with those of other studies concerning alcohol addiction, i.e., those who consume alcohol in harmful ways display lower resilience levels than the general population [54]. Similarly, as previously shown, non-smoking nurses in China had significantly higher resilience levels than those who smoked [46]. We also found that nurses with high resilience had better eating habits, which is in line with other reports that show people with compulsive overeating habits have low levels of resilience [55]. Likewise, a study conducted among young people showed that high resilience was a protective factor against risky forms of behaviour, such as smoking, alcohol consumption, and taking drugs [56].

As regards the conducted studies, personality resilience should be regarded as a significant factor influencing some aspects of lifestyle among Polish nurses. Therefore, the authors aim to present the results of their studies to nursing managers and to representatives of teaching hospitals. The aforementioned results will form the basis for the preparation of educational programmes. Considering the increasing demand for care and ageing of nursing personnel, we must consider the implementation of strategies aimed at the improvement of their psychophysical mood. Taking the above into consideration, workplace managers should implement strategies aimed at evaluating the psychophysical mood and mental resilience of personnel, e.g., via the application of screening questionnaires that can identify at-risk personnel. The suggested evaluation could be applied, for example, during periodic medical examinations undertaken in the workplace. It is important for employers to take care not only of the physical health of their employees but also their mental health. It would be advisable to offer psychological assistance to those personnel who require it. The conducted research indicates that an evaluation of mental resilience should be created particularly for younger or less experienced nurses. Resilience as an inborn energy or vital force may enhance the position of nurses by positively adjusting them to stressful situations and may apply their experiences as a learning process [57]. Resilience may enhance the ability of nurses to effectively cope with stress (thanks to a better psychophysical mood), as well as reduce unhealthy forms of behaviour (habitual or emotional overeating, the application of dietary restrictions, and the consumption of alcohol in a risky way). Relatedly, hospital administrators and managers of nursing departments should realise the importance of mental resilience and regularly evaluate the issue of nurses’ mental health. The need for continuous supervision over the preparation of nurses to perform their professional duties was also mentioned in another Polish study [58]. What seems to be advisable as well is the institution of coherent measurement tools which could allow researchers to compare diverse research results aiming at drawing the relevant conclusions related to the improvement of nurses’ resilience. It seems that the tools used in the study properly reviewed the areas considered.

Our study had some limitations. The data collected in the study were based on voluntary questionnaires carried out in one region of Poland. Therefore, the findings may not reflect all nurses working in all regions of Poland. However, we provided a large cohort, which ensures that the data are somewhat representative of the national population of nurses in Poland. Self-descriptive measurement tools were used in the studies, which are associated with the possibility of the occurrence of a variable social approval, i.e., the will of respondents to be presented in a better light. Moreover, the studies were cross-sectional. Subsequently, it was not possible to unequivocally formulate cause-and-effect relations on their basis.

## 5. Conclusions

A significant percentage of nurses in Poland manifested average and low mental resilience. This was associated with unhealthy lifestyle behaviour and deteriorating mental and physical condition, which could adversely affect their professional performance and increase the risk of chronic diseases. Particular attention should also be paid to younger nurses who show lower mental resilience, which, in the absence of any intervention, may result in the deterioration of one’s mental and physical condition and the occurrence of risky behaviour. Moreover, both mental resilience and good physical condition were associated with additional employment. Therefore, taking into account mental factors such as personality resilience in preventive examinations will improve the professional performance of nurses.

## Figures and Tables

**Table 1 ijerph-18-01807-t001:** General characteristics of the analysed group of nurses in Poland.

Characteristic	*N* (%)
**Sex**	
Male	28 (2.6%)
Female	1052 (97.4%)
**Age (years)**	
≤30	132 (12.3%)
31–40	208 (19.2%)
41–50	587 (54.3%)
≥51	153 (14.2%)
**Marital status**	
Single	760 (71.1%)
Married	105 (9.8%)
Divorced	178 (16.7%)
Widowed	26 (2.4%)
**Additional employment**	
Yes	397 (36.8%)
No	638 (63.2%)
**Type of ward**	
Surgical	354 (32.7%)
Nonsurgical	726 (67.3%)
**Attitude to smoking** ^†^	
Smoker	214 (19.8%)
Ex-smoker	182 (16.9%)
Non-smoker	684 (63.3%)
**Alcohol consumption** ^‡^	
Risk of alcohol addiction	378 (35%)
No risk of alcohol addiction	702 (65%)
**Psychophysical mood** ^¶^	
High	166 (15.4%)
Average	735 (68%)
Low	179 (16.6%)
	**M (SD)**
Psychophysical mood	3.56 (0.51)
Physical wellbeing (D1)	3.68 (0.59)
Mental mood (D2)	3.45 (0.53)
**Resiliency** ^⁜^	
High	176 (16.3%)
Average	649 (60.1%)
Low	255 (23.6%)
	**M (SD)**
Determination and persistence in action	14.26 (3.23)
Openness to new experiences and a sense of humour	13.87 (3.13)
Personal competencies to cope and tolerance of negative effect	13.40 (3.40)
Tolerance of failures and treating life as a challenge	13.89 (3.76)
Optimistic life attitude and ability to mobilise in difficult situations	12.87 (3.33)
**Eating habits** ^§^	**M (SD)**
Eating habits—total	10.65 (5.96)
Restraint from eating	3.56 (2.24)
Emotional overeating	4.10 (2.59)
Habit overeating	3.00 (2.62)

^†^ Based on the Fagerström test for nicotine dependence. ^‡^ Based on the Alcohol Use Disorders Identification Test for Consumption (AUDIT-C) screening test for risk of alcohol abuse. ^¶^ Based on the Psychosocial Working Conditions (PWP) questionnaire. ^⁜^ Based on the Assessment of Resiliency Scale (SPP-25). ^§^ Based on My Eating Habits (MEH) questionnaire. Note: Any questionnaires that were completed incorrectly were excluded from the analysis; therefore, numbers may not add up to 1080. M, mean.

**Table 2 ijerph-18-01807-t002:** Characteristics of the studied group of nurses, including psychophysical wellbeing and sociodemographic features.

Wellbeing Scale ^#^	Age (Years)					
	≤30	31–40	41–50	≥51	Kruskal–Wallis ANOVA	
	M (SD)	M (SD)	M (SD)	M (SD)	*p*-Value	
Psychophysical mood	3.59 (0.46)	3.62 (0.58)	3.54 (0.50)	3.56 (0.51)	0.302	
Physical wellbeing (D1)	3.74 (0.55)	3.72 (0.64)	3.66 (0.59)	3.68 (0.57)	0.210	
Mental mood (D2)	3.43 (0.47)	3.51 (0.60)	3.43 (0.52)	3.44 (0.55)	0.161	
	**Marital status**					
	**Single**	**Married**	**Divorced**	**Widowed**	**(One-way ANOVA)**	
	**M (SD)**	**M (SD)**	**M (SD)**	**M (SD)**	***p*-value**	
Psychophysical mood	3.61 (0.48)	3.58 (0.52)	3.52 (0.56)	3.31 (0.64)	0.028	
Physical wellbeing (D1)	3.77 (0.55)	3.69 (0.59)	3.67 (0.60)	3.35 (0.87)	0.009	
Mental mood (D2)	3.46 (0.51)	3.47 (0.53)	3.37 (0.61)	3.26 (0.59)	0.079	
	**Additional employment**					
	**Yes**		**No**		**(Mann–Whitney U Test)**	
	**M (SD)**		**M (SD)**		***Z***	***p*-value**
Psychophysical mood	3.62 (0.52)		3.54 (0.52)		−2.395	0.017
Physical wellbeing (D1)	3.74 (0.58)		3.66 (0.59)		−2.062	0.039
Mental mood (D2)	3.50 (0.59)		3.42 (0.53)		−2.475	0.013
	**Type of ward**					
	**Surgical**		**Non-surgical**		**(Mann–Whitney U Test)**	
	**M (SD)**		**M (SD)**		***Z***	***p*-value**
Psychophysical mood	3.58 (0.52)		3.56 (0.52)		−0.571	0.568
Physical wellbeing (D1)	3.71 (0.59)		3.68 (0.59)		−0.727	0.467
Mental mood (D2)	3.46 (0.52)		3.44 (0.54)		−0.301	0.763

Abbreviation: *M* (*SD*), mean (standard deviation). ^#^ Based on the Psychosocial Working Conditions (PWP) questionnaire.

**Table 3 ijerph-18-01807-t003:** Correlations between the wellbeing of Polish nurses and their smoking, drinking, and eating habits.

Parameter	Psychophysical Mood	Physical Wellbeing (D1)	Mental Mood (D2)
	*r* (*p*-Value)	*r* (*p*-Value)	*r* (*p*-Value)
Fagerström test for nicotine ^†^	−0.079 (0.010)	−0.074 (0.015)	−0.073 (0.017)
AUDIT-C test ^‡^	−0.096 (0.002)	−0.080 (0.009)	−0.096 (0.002)
Eating habits—total ^#^	−0.234 (<0.001)	−0.209 (<0.001)	−0.220 (<0.001)
Restraint from eating ^#^	−0.126 (<0.001)	−0.123 (<0.001)	−0.105 (<0.001)
Emotional overeating ^#^	−0.232 (<0.001)	−0.206 (0.001)	−0.219 (<0.001)
Habit overeating ^#^	−0.161 (<0.001)	−0.130 (<0.001)	−0.168 (<0.001)

Abbreviation: *r*, Spearman’s coefficients of rank correlation. ^†^ Based on the Fagerström test for nicotine dependence. ^‡^ Based on the AUDIT-C screening test for risk of alcohol abuse. ^#^ Based on My Eating Habits (MEH) questionnaire.

**Table 4 ijerph-18-01807-t004:** Relationships between resilience and the sociodemographic characteristics of Polish nurses.

Parameter		^⁜^ Factors				
	Resiliency—Total	1	2	3	4	5
	M (SD)	M (SD)	M (SD)	M (SD)	M (SD)	M (SD)
Age (years)						
≤30	64.67 (14.37)	13.51 (3.38)	13.31 (3.15)	12.70 (3.25)	13.03 (3.14)	12.12 (3.16)
31–40	69.54 (14.54)	14.53 (3.04)	14.28 (3.11)	13.62 (3.34)	14.07 (3.11)	13.04 (3.36)
41–50	68.37 (14.70)	14.29 (3.13)	13.85 (3.07)	13.44 (3.37)	13.88 (3.25)	12.91(3.29)
≥51	68.72 (15.88)	14.29 (3.15)	13.86 (3.33)	13.56 (3.68)	13.92 (3.59)	13.08 (3.56)
*p*-Value (Kruskal–Wallis ANOVA)	0.004	0.017	0.024	0.017	0.011	0.016
Marital status						
Single	67.52 (14.99)	13.99 (3.23)	13.99 (3.09)	13.26 (3.32)	13.63 (3.30)	12.64 (3.42)
Married	68.66 (14.50)	14.35 (3.13)	13.91 (3.07)	13.50 (3.33)	13.90 (3.21)	12.99 (3.21)
Divorced	68.06 (15.68)	14.42 (2.86)	13.71 (3.28)	13.32 (3.82)	13.90 (3.54)	12.70 (3.71)
Widowed	63.27 (19.88)	12.77 (4.38)	12.85 (4.26)	12.85 (4.14)	12.88 (3.94)	11.92 (4.59)
*p*-Value (One-way ANOVA)	0.268	0.048	0.328	0.656	0.355	0.236
Additional employment						
Yes	69.46 (14.49)	14.38 (3.11)	14.03 (3.11)	13.84 (3.38)	14.11 (3.22)	13.11 (3.16)
No	67.46 (15.01)	14.16 (3.19)	13.78 (3.15)	13.14 (3.39)	13.65 (3.29)	12.73 (3.43)
*p*-Value (Mann–Whitney U Test)	0.008	0.109	0.157	0.01	0.015	0.048
*Z*	−2.656	−1.604	−1.416	3.954	−2.423	−1.975
Type of ward						
Surgical	67.65 (14.47)	14.07 (3.15)	13.76 (3.11)	13.25 (3.34)	13.84 (3.89)	12.84 (3.23)
Non-surgical	68.46 (15.03)	14.35 (3.26)	13.92 (3.15)	13.47 (3.43)	13.92 (3.70)	12.88 (3.39)
*p*-Value (Mann–Whitney U Test)	0.205	0.202	0.287	0.167	0.287	0.843
*Z*	−1.268	−1.276	−1.066	−1.382	−1.065	−0.198

Abbreviation: *M* (*SD*), mean (standard deviation). ^⁜^ Based on the Assessment of Resiliency Scale (SPP-25). 1. Determination and persistence in action. 2. Openness to new experiences and a sense of humour. 3. Personal competencies to cope and tolerance of negative effect. 4. Tolerance of failures and treating life as a challenge. 5. Optimistic life attitude and ability to mobilise in difficult situations.

**Table 5 ijerph-18-01807-t005:** Comparison of resilience among various forms of risky behaviour and wellbeing.

Parameter	Resiliency Low ^#^ (*n* = 255) *N* (%)	Resiliency Average ^#^ (*n* = 649) *N* (%)	Resiliency High ^#^ (*n* = 176) *N* (%)	*p*-Value (Chi-Squared Test)
Attitude to smoking				0.711
Smokers	39 (15.3%)	109 (16.8%)	32 (18.13%)	
Non-smokers	216 (84.7%)	540 (83.2%)	143 (81.7%)	
Alcohol consumption ^†^				0.003
Risk of alcohol addiction	95 (37.3%)	241 (37.1%)	42 (23.9%)	
No risk of alcohol addiction	160 (62.7%)	408 (62.9%)	134 (76.1%)	
	**M (SD)**	**M (SD)**	**M (SD)**	***p*-value** **(Kruskal–Wallis ANOVA)**
Wellbeing ^¶^				
Psychophysical mood	3.28 (0.53)	3.62 (0.46)	3.80 (0.53)	<0.001
Physical wellbeing	3.46 (0.58)	3.73 (0.56)	3.85 (0.64)	<0.001
Mental mood	3.11 (0.56)	3.51 (0.46)	3.74 (0.52)	<0.001
Eating habits ^§^				
Eating habits—total	12.01 (6.68)	10.35 (5.66)	9.80 (5.62)	0.001
Restraint from eating	3.56 (2.25)	3.59 (2.25)	3.42 (2.22)	0.711
Emotional overeating	4.64 (2.86)	3.97 (2.47)	3.79 (2.54)	0.002
Habit overeating	3.80 (2.96)	2.79 (2.47)	2.59 (2.36)	<0.001

Abbreviation: *M* (*SD*), mean (standard deviation). ^#^ Based on the Assessment of Resiliency Scale (SPP-25). ^†^ Based on the AUDIT-C screening test. ^¶^ Based on the Psychosocial Working Conditions (PWP) questionnaire. ^§^ Based on My Eating Habits (MEH) questionnaire. Note: Any questionnaires that were completed incorrectly were excluded from the analysis; therefore, numbers may not add up to 1080.

## Data Availability

Not applicable.

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
