# Peer review of "Risky Behaviour among Nurses in Poland: An Analysis of Nurses’ Physical Condition, Mental Health, and Resilience"

_ijerph, 2021, doi:10.3390/ijerph18041807_

Round 1

Reviewer 1 Report

Please, see the file attached.

Reviewer 2 Report

A study dedicated on nurses’ analysis of mental and physical condition and mental resilience in Poland

Nurses are the most stable component in the emergency system.  While physicians come and go, it is the experienced nurses who have been there for years to “remind” us physicians what should be done when crisis’s strikes. 

The manuscript submitted by Lucyna Gieniusz-Wojczyk evaluated the information about nurses working in Poland.  The study based on a survey reveals nurses with average and low mental resilience, which is unfavorable to health, constituted a significant part of the group being the object of the analysis.

Please consider changing the title.  Word “Occurrence” doesn’t sounds good in the tittle, especially at the beginning. My suggesition: “Risky Behaviours Among Nurses in Poland: The Physical Condition, Mental Health and Resilience”

The necessity and innovation of the article should be presented to the introduction.

The most important contribution of both studies in my opinion is the survey form they produced.  The survey should be of value for the readers of this journal who wish to evaluate the preparedness in the areas where they serve. As much as this survey is concerned: The authors should describe the full scale of key words used to construct the survey.

Now that the authors have accumulated experience with 1080 responders, they should add a section in the discussion section evaluating the survey questions they constructed explaining if and how the questions should be modified.

The authors should explain how the nurses to which the survey was sent were chosen.

Information concerning the study population should describe how many nurses the survey was sent to.  This way we can realize the response rate.  As written, one understands that the response rate was 100%.

The observation that assessing workplace as prepared is associated with one’s perception of being prepared is both logical and problematic in an environment.  This should be discussed in the limitations section.

The authors should decide if they want to add a note or not on whether the results of this survey were presented or not to local health authorities. Though not an integral part of research, we have an ethical responsibility to disseminate findings that might have an impact on local preparedness.

It is suggested to compare the results of the present research with some similar studies which is done before.

Please make sure your conclusions' section underscore the scientific value added of your paper, and/or the applicability of your findings/results, as indicated previously. Please revise your conclusion part into more details. Basically, you should enhance your contributions, limitations, underscore the scientific value added of your paper, and/or the applicability of your findings/results and future study in this session.

I highly recommend more precise literature research and more carefully writing

Some additional copy editing, both to smooth out the areas of clumsy prose due to translation and to use translated terms that are consistent with current medical vernacular will also be required

Reviewer 3 Report

Firstly, I would like to thank the authors for the opportunity of reading and reviewing their manuscript. The research is well conducted. The paper is a descriptive study and basically presents the results of a survey-based research among Polish nurses. The research is well designed and well-conducted, and the results are properly presented.
I have some concerns/suggestions:
- Regarding the theoretical framework, I think it would be useful to apply a theoretical model or theory that relates resilience with exposure to stress and the other variables analyzed. This would help to better explain the results obtained and would add more value to the discussion section.
- About materials and methods, the psychometric properties of the scales should be specified.
- Regarding the references, it is convenient that they are reviewed. Some are incomplete, and many of them do not have the doi.
Good luck!

Round 2

Reviewer 2 Report

Authors referred to all my suggestions and also added a lot of new information suggested by other reviewers.
I believe, in this form the manuscript is definitely stronger.
Just few minor suggestions:

Vocabulary needs to be standardized, sometimes it's British and sometimes American.
Overall, there are some typos and errors i.e.

line 66: on the basis of -> based on
line 93: should be 'to overeat'

Some new important study appeared from Poland, which in my opinion is worth to refer. Not directly related to mental health, but points and show importance of of nurses in the emergency system. https://doi.org/10.1371/journal.pone.0244488

Congratulations to the authors for their contributions.
I am very much looking forward to reading your next paper!

Reviewer 3 Report

Dear authors,

Thanks for your reply. The explanations of the authors are satisfactory. The paper has greatly improved its quality.

Congratulations on your work.

Author Response

Dear Reviewer,

Thank you for your positive review of our manuscript (ijerph-1055643): Risky Behaviour Among Nurses in Poland: An Analysis of Nurses' Physical Condition, Mental Health and Resilience.

Sincerely,

Lucyna Gieniusz-Wojczyk